# Effects of supplementing a polyphenol-rich sugarcane extract through drinking water on egg production and quality of laying hens

Namalika D. Karunaratne[1]*, Sasmitha De Silva[1], Minoli Herath[1], Ruvini Liyanage[2], Pabodha Weththasinghe[3], Barana C. Jayawardana[3], Eranga De Seram[1], Anil Pushpakumara[1], Mathew Flavel[4]

**1** Department of Farm Animal Production and Health, Faculty of Veterinary Medicine and Animal Science, University of Peradeniya, Peradeniya, Sri Lanka, **2** National Institute of Fundamental Studies, Kandy, Sri Lanka, **3** Department of Animal Science, Faculty of Agriculture, University of Peradeniya, Peradeniya, Sri Lanka, **4** Bioactives Division, The Product Makers, Melbourne, Victoria, Australia

\* nkarunaratne@vet.pdn.ac.lk

## Abstract

Polyphenols are a wide group of naturally occurring compounds found in plants and have the potential to safeguard living cells. The objective was to evaluate whether the inclusion of a polyphenol-rich sugarcane extract (PRSE) in drinking water could improve egg production and the quality of commercial layers. A total of 120 Shaver Brown hens, aged 43 weeks, were randomly allocated to 12 litter-floor pens in two open-sided poultry houses. The pens were divided into two treatment groups: one receiving 0% (control) and the other 0.05% PRSE in drinking water throughout the study duration. The treatments were prepared by adding PRSE manually into the drinking water daily, and water was given *ad libitum*. The birds were given commercial layer feed throughout the study. The number of eggs produced, abnormal eggs, and mortality were recorded daily. Egg weight, yolk colour, yolk height, albumen height, Haugh units, and antioxidant properties, were measured at weeks 45, 47 and 49. Supplementing PRSE in the drinking water did not impact hen-day egg production, hen-housed egg production, egg weight, egg mass, or feed conversion ratio. However, there was a trend toward significance in egg weight at week 45. The results indicated that PRSE supplementation led to a significant reduction in yolk colour during week 45 (P = 0.001), although no differences were observed in subsequent weeks. Yolk height, thick albumen height, and haugh units were unaffected by the treatment, while thin albumen height showed a trend towards reduction in the PRSE group at weeks 47 and 49 (P = 0.05). The DPPH assay revealed a significant increase in antioxidant capacity in the PRSE group at week 45 (P = 0.02). The 0.05% PRSE supplementation in drinking water initially enhanced antioxidant capacity but later adversely affected yolk color and thin albumen height.

## Introduction

The poultry industry plays a critical role in meeting the global demand for high-quality protein, particularly through the production of meat and eggs. With advancements in

**Data availability statement:** All relevant data are included within the manuscript and its Supporting Information files (S1 Dataset).

**Funding:** This research was funded by The Produce Makers, VIC, Australia. The funders did not inappropriately influence the study design, data collection, analysis and interpretation, decision to publish, or preparation of the manuscript.

**Competing interests:** Mathew Flavel, a co-author of this manuscript is attached to the company that supplies the polyphenol product. However, he did not involve with manuscript writing, data analysis and interpretation.

genetics, nutrition, and management practices, the poultry sector has achieved substantial gains in productivity [1]. However, challenges such as environmental stress, disease, and feed costs continue to pose significant hurdles, necessitating innovative strategies to enhance poultry health, performance, and sustainability [2]. Among these challenges, oxidative stress has emerged as a critical factor impacting the overall productivity and health of poultry [3].

The poultry industry is increasingly challenged by oxidative stress, which arises from various factors such as mycotoxins and heat stress, leading to lipid peroxidation and cellular apoptosis [4]. This oxidative stress is detrimental to poultry health, negatively impacting performance and liveability by suppressing the immune system [5]. Consequently, mitigating oxidative stress through the incorporation of antioxidants, particularly natural ones, has become a focal point in enhancing poultry health and productivity [6].

In recent years, the integration of natural bioactive compounds into poultry diets, especially for laying hens, has garnered substantial attention within the field of poultry nutrition. Among these bioactive compounds, polyphenols, which are naturally occurring substances found in plants are of particular interest due to their wide range of biological activities, including antioxidant, anti-inflammatory, anticancer, and antimicrobial properties [7,8]. These compounds include a diverse array of chemical structures, such as phenolic acids, flavonoids, and tannins, which contribute to their functional versatility [9]. Sugarcane (*Saccharum officinarum*), a perennial grass commonly found in South Asia, Southeast Asia, and other tropical and subtropical regions, is notably rich in polyphenolic compounds, such as phenolic acids, glycosides, flavonoids, and fatty acids [10,11]. Additionally, sugarcane is rich in policosanols, and steroids, which makes sugarcane superior to other sources of plant-derived polyphenols [11]. Therefore, sugarcane can be used as a significant source of plant-derived polyphenols to address oxidative stress in poultry.

These polyphenolic compounds have been shown to positively influence various physiological processes in poultry, contributing to improved gut health, enhanced growth performance, reduced lipid peroxidation, and the mitigation of oxidative stress. Moreover, polyphenols are integral to plant defense mechanisms, and when incorporated into livestock diets, they offer numerous health benefits [12,13]. Notably, phenolic compounds derived from plant materials, such as sugarcane extracts, have demonstrated anticoccidial effects in poultry [14]. Further, it has been observed that sugarcane extracts improve broiler performance, including weight gain and feed conversion ratio [7]. Therefore, understanding the composition and properties of polyphenols in sugarcane is essential for advancing agricultural practices and animal-derived food production.

Despite the extensive body of knowledge regarding the role of polyphenols in livestock nutrition, there remains a significant research gap concerning the specific impact of polyphenol-rich sugarcane extract (PRSE) on egg production and quality in laying hens. While previous studies have examined polyphenols from various plant sources, the unique composition of sugarcane-derived polyphenols and their potential effects on layer production have not been thoroughly investigated. Moreover, the limited studies that are available on this topic have yielded inconsistent results, emphasizing the need for further research [15].

In response to this research gap, the present study was designed to assess the impact of PRSE supplementation in drinking water on the egg production and quality of laying hens. It was hypothesized that PRSE administration in drinking water would enhance both egg production and quality in laying hens. This investigation aims to provide a more comprehensive understanding of the effects of polyphenol-rich sugarcane extracts on poultry performance, thereby contributing valuable insights to the field of poultry nutrition.

## Materials and methods

The experimental protocol received approval from the Ethics Committee of the Faculty of Veterinary Medicine and Animal Science at the University of Peradeniya, as indicated by the Ethical Clearance Certificate Number VERC-24-02.

### Experimental treatments

Two sets of treatments were formulated by manually blending 0% and 0.05% of PRSE in drinking water individually. Drinking water with or without PRSE was provided as *ad libitum* throughout the duration of the study (from week 43 to 49 of age).

### Birds and housing

A total of 120 Shaver Brown laying hens, aged 43 weeks, were randomly allocated to 12 floor pens, with 10 birds per pen. Each pen contained paddy husk evenly spread across the concrete floor. These pens were located in two open-sided poultry houses (6.2 m in length, 6 m in width, 2.2 m in height) at the Udaperadeniya Livestock Farm, affiliated with the Faculty of Agriculture, University of Peradeniya, Sri Lanka. Each poultry house was divided into six pens (1.8 m in length, 1.8 m in width, 1.7 m in height). The two treatments were randomly assigned to three pens in each house, resulting in six replications per treatment. Each pen was equipped with a manual plastic pan feeder (10 kg capacity; 40 cm pan diameter) and a manual plastic bell drinker (6 l capacity; 23 cm diameter). Additionally, each pen contained three nest boxes (30.5 cm in length, 30.5 cm in width, 28 cm in height). The laying hens were provided with commercial layer feed in mash form throughout the study period. Each pen received 1.2 kg of feed per day, corresponding to 120 g of feed per bird per day. The environmental and management conditions were maintained according to the recommended standards of the Shaver Brown management guidelines. The temperature and humidity could not be controlled during the trial as the study was conducted in an open-sided poultry house. Environmental temperatures ranged from 22 to 32°C, while humidity levels fluctuated between 70% and 85%.

### Study material

In this study, a recently identified sugarcane extract was employed, which was initially developed using the hydrophobic resin method described by [8]. The extract, available in a liquid-to-paste form and exhibiting a brown coloration, was produced by The Product Makers Pvt Ltd in Melbourne, Australia. The sugarcane extract contained a total polyphenol content of 221 mg/g and a flavonoid content of 53.8 mg/g, and the pH of a 1% solution at 20°C was measured at 3.74 [16]. Furthermore, the extract demonstrated a total antioxidant activity of 18,837 µmol/g. Elemental analysis showed concentrations of sodium (Na), potassium (K), calcium (Ca), magnesium (Mg), and zinc (Zn) at 170, 130, 6600, 3000, and 15 mg/kg, respectively, with selenium (Se) and chromium (Cr) present at 0.3 and 2.4 mg/kg [8].

### Data collection

The total number of eggs produced and the number of abnormal eggs (including broken, cracked shell, soft shell, abnormal shaped, and double-yolked eggs) were recorded daily for each pen throughout the study period. Hen-day egg production and hen-housed egg production were calculated by dividing the total number of eggs per pen by the number of hen-days and hens housed, respectively [17]. Eggs laid in each pen over a 24-hour period were collected and individually weighed using a digital scale (SF-400, LK) at 45, 47 and 49 weeks of age, allowing for the calculation of average egg weight per pen on these days. Egg

mass was determined by multiplying the total number of eggs by the average egg weight on the specified days [18]. Mortality was recorded daily, and the feed conversion ratio was calculated biweekly (at 45, 47 and 49 weeks of age) as g of feed consumed per g of egg mass [17]. Humane endpoints during the research were determined based on several criteria, including significant impairment of the bird's mobility that affects its capacity to access food and water, substantial weight loss, and the presence of a severe defect with no possibility of recovery.

## Determination of egg quality

Two of the collected eggs from each pen at 45, 47, and 49 weeks of age were used to analyze thin albumen height, thick albumen height, yolk height, and yolk colour. Immediately after collection, eggs were transported to a laboratory and analyzed the same day. Thick and thin albumin height, yolk height, and yolk color values used in this study were the mean of three measurements obtained directly from each egg after breaking it on a glass surface. Eggs were measured with an electronic scale (SF-400, LK) before breaking the egg. The heights of the thick and thin albumin and yolk were measured using a digital Haugh meter (Model S6428, b.c. Ames Co., Waltham, Mass, USA), and a yolk colour fan (ROCHE, Yolk Colour fan, 1155, Switzerland) was used to measure yolk colour. Three readings from different places were taken and the mean value was used for the calculations.

Haugh Unit (HU) was calculated according to the formula below.

$$HU = 100 * \log_{10}\left(h - 1.7w^{0.37} + 7.56\right)$$

Observed mean height of the thick albumen in millimetres (h) and mean value of the weight of eggs per pen in grams (w) were used for calculations [19].

## Determination of antioxidant properties of eggs

Five eggs collected from the same pen were carefully cracked, and the yolks were separated using an egg yolk separator. The yolks from all five eggs were pooled together and thoroughly mixed using a hand mixer. A 60 ml aliquot of the pooled egg yolk mixture was then freeze-dried using a freeze dryer (SUPERMODULYO-230, Milford, MA 01757) and stored at −20°C until further analysis. The resulting freeze-dried egg yolk powder was utilized for subsequent sample extraction. Powdered egg yolk samples (2 g) were extracted into 20 ml of 80% methanol (analytical grade) by magnetic stirring for 30 minutes. Then the content was centrifuged for 15 minutes at 6000 rpm at 4°C using a centrifuge (5340R, Germany) and the supernatant was separated and stored at −20°C until analysis.

**Determination of free radical scavenging capacity using 2,2-Diphenyl-1-Picrylhydrazyl (DPPH) assay.** The procedure described by [20] was used with minor modifications. A hundred microliters of DPPH solution were added to different extract volumes (0–150 l). The reaction mixture was allowed to stand in the dark for 30 min at room temperature, and the absorbance was measured at 517 nm. The results were expressed in mmol/dm$^3$ Trolox equivalents per gram dry weight.

**Determination of ferric reducing ability using Ferric Reducing Antioxidant Power (FRAP) assay.** The FRAP reagent (150 μl) containing 2,4,6-Tripyridyl-S-triazine (TPTZ) (10 mM in 10 mM HCl), $FeCl_3$ (10 mM), and 30 μl of pH 3.6 acetate buffer (300 mM) with the ratio of 1:1:10 (v/v/v) was pre-incubated at 37°C for 8 min. The sample extract (50 μl) was added to 150 μl of the FRAP reagent and incubated for 30 min at room temperature. The

absorbance was measured at 593 nm, and results were expressed in $\mu mol dm^{-3}$ $Fe^{2+}$ equivalents per gram dry weight.

## Statistical analysis

The study employed a randomized complete block design, utilizing a broiler room as a block to alleviate environmental impacts. Egg production and quality, and antioxidant properties were analyzed through a two-sample T-test using the Proc mixed model in the Stata statistical software. The significance level was set at $P \leq 0.05$. Before variance analysis, data normality was assessed through the Shapiro-Wilk test.

## Results

### Egg production

Supplementation of PRSE in drinking water revealed no significant impact on egg production, egg weight, egg mass and feed conversion ratio of laying hens across each evaluated period or when considering the total duration of the study as detailed in Tables 1 and 2. Weekly hen-day egg production and hen-housed egg production were illustrated in Figs 1 and 2, respectively. The egg weight at week 2 tended to be higher in PRSE supplemented group compared to the control (P = 0.06).

**Table 1. Effect of supplementing polyphenol-rich sugarcane extract (PRSE) in drinking water on the egg production of laying hens.**

| Variable | PRSE[1] concentration (%) in drinking water | | SEM[2] | P value |
|---|---|---|---|---|
| | 0 | 0.05 | | |
| Weekly hen-day egg production (%) | | | | |
| 44 wk of age | 91.4 | 88.8 | 2.696 | 0.35 |
| 45 wk of age | 94.3 | 94.3 | 2.857 | 1.00 |
| 46 wk of age | 91.9 | 95.5 | 3.280 | 0.30 |
| 47 wk of age | 89.5 | 93.1 | 5.124 | 0.49 |
| 48 wk of age | 93.4 | 91.7 | 2.591 | 0.51 |
| 49 wk of age | 95.8 | 92.9 | 3.216 | 0.37 |
| Bi-weekly hen-day egg production (%) | | | | |
| 43 to 45 wk of age | 92.9 | 91.5 | 2.558 | 0.62 |
| 45 to 47 wk of age | 90.7 | 94.3 | 3.946 | 0.39 |
| 47 to 49 wk of age | 94.6 | 92.3 | 2.506 | 0.37 |
| Total hen-day egg production (%) | 92.7 | 92.7 | 2.397 | 0.99 |
| Weekly hen-housed egg production (%) | | | | |
| 44 wk of age | 91.4 | 88.8 | 2.696 | 0.35 |
| 45 wk of age | 94.3 | 94.3 | 2.857 | 1.00 |
| 46 wk of age | 91.9 | 95.5 | 3.280 | 0.30 |
| 47 wk of age | 88.8 | 93.1 | 5.628 | 0.46 |
| 48 wk of age | 91.9 | 91.7 | 3.280 | 0.94 |
| 49 wk of age | 94.3 | 92.9 | 3.762 | 0.71 |
| Bi-weekly hen-housed egg production (%) | | | | |
| 43 to 45 wk of age | 92.9 | 91.5 | 2.558 | 0.62 |
| 45 to 47 wk of age | 90.4 | 94.3 | 4.196 | 0.37 |
| 47 to 49 wk of age | 93.1 | 92.3 | 3.195 | 0.80 |
| Total hen-housed egg production (%) | 92.1 | 92.7 | 2.746 | 0.83 |

[1]PRSE—Polyphenol rich sugarcane extract.

[2]SEM—Pooled standard error of the mean (n = 6).

**Table 2. Effect of supplementing polyphenol-rich sugarcane extract (PRSE) in drinking water on the production performance of laying hens.**

| Variable | PRSE[1] concentration (%) in drinking water | | SEM[2] | P value |
|---|---|---|---|---|
| | 0 | 0.05 | | |
| Egg weight (g) | | | | |
| 45 wk of age | 59.8 | 61.5 | 0.831 | 0.06 |
| 47 wk of age | 60.4 | 61.1 | 1.005 | 0.48 |
| 49 wk of age | 61.6 | 61.2 | 0.938 | 0.68 |
| Egg mass (g/hen/day) | | | | |
| 45 wk of age | 56.7 | 58.5 | 4.228 | 0.67 |
| 47 wk of age | 53.1 | 54.1 | 3.353 | 0.76 |
| 49 wk of age | 51.2 | 54.1 | 5.120 | 0.59 |
| Feed conversion | | | | |
| 45 wk of age | 2.15 | 2.08 | 0.155 | 0.67 |
| 47 wk of age | 2.28 | 2.25 | 0.155 | 0.89 |
| 49 wk of age | 2.48 | 2.23 | 0.322 | 0.45 |

[1]PRSE—Polyphenol rich sugarcane extract.

[2]SEM—Pooled standard error of the mean (n = 6).

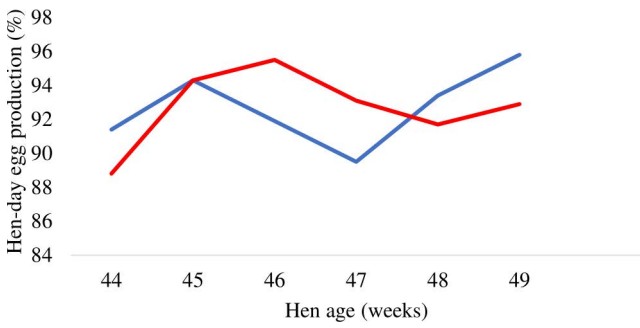

**Fig 1. Effect of supplementing polyphenol-rich sugarcane extract (PRSE) in drinking water on the weekly hen-day egg production of laying hens.** The blue line denotes 0% PRSE supplementation and the red line indicates 0.05% PRSE supplementation.

## Egg quality

At 45 weeks, egg yolk color was reduced with PRSE supplementation compared to the control group (Table 3). However, no significant differences in yolk color were observed at 47 and 49 weeks of age between the treatments. Polyphenol-rich sugarcane extract supplementation led to a reduction in thin albumen height at the evaluated time points, except at 45 weeks. No significant treatment effects were observed on yolk height, thick albumen height, or Haugh units across all weeks of age analyzed (Table 3).

## Antioxidant properties

Supplementation of PRSE in drinking water reduced the 2,2-diphenyl-1-picrylhydrazyl (DPPH) assay value, indicating an enhanced free radical scavenging capacity in laying hens at 45 weeks of age (Table 4). However, supplementation of PRSE in drinking water demonstrated no significant effect on reducing ferric ion to ferrous ion in laying hens across each evaluation period (P > 0.05), as detailed in Table 4.

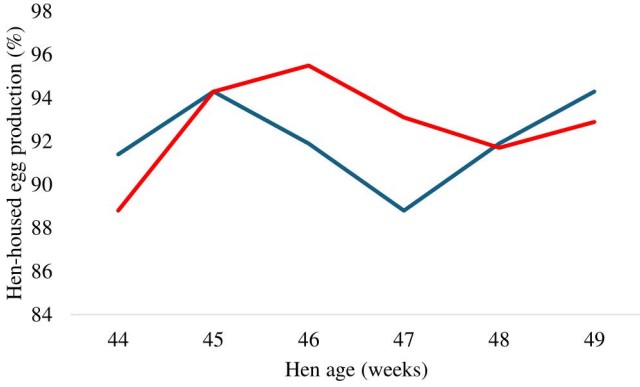

**Fig 2. Effect of supplementing polyphenol-rich sugarcane extract (PRSE) in drinking water on the weekly hen-housed egg production of laying hens.** The blue line denotes 0% PRSE supplementation and the red line indicates 0.05% PRSE supplementation.

**Table 3. Effect of supplementing polyphenol-rich sugarcane extract (PRSE) in drinking water on egg quality.**

| Variable | PRSE[1] concentration (%) in drinking water | | SEM[2] | P value |
|---|---|---|---|---|
| | 0 | 0.05 | | |
| Yolk colour | | | | |
| 45 wk of age | 7.3[a] | 6.6[b] | 0.120 | 0.001 |
| 47 wk of age | 6.7 | 6.7 | 0.112 | 1.00 |
| 49 wk of age | 6.3 | 6.7 | 0.151 | 0.29 |
| Yolk height (mm) | | | | |
| 45 wk of age | 17.4 | 17.5 | 0.537 | 0.87 |
| 47 wk of age | 17.8 | 18.0 | 0.256 | 0.87 |
| 49 wk of age | 18.4 | 18.8 | 0.161 | 0.24 |
| Thick albumen height (mm) | | | | |
| 45 wk of age | 7.7 | 7.6 | 0.206 | 0.96 |
| 47 wk of age | 8.0 | 8.0 | 0.341 | 0.99 |
| 49 wk of age | 8.4 | 8.7 | 0.196 | 0.47 |
| Thin albumen height (mm) | | | | |
| 45 wk of age | 2.0 | 1.9 | 0.104 | 0.66 |
| 47 wk of age | 2.6[a] | 2.0[b] | 0.171 | 0.05 |
| 49 wk of age | 3.1[a] | 2.7[b] | 0.111 | 0.05 |
| Haugh units | | | | |
| 45 wk of age | 87.3 | 87.0 | 1.237 | 0.14 |
| 47 wk of age | 89.2 | 88.9 | 1.786 | 0.62 |
| 49 wk of age | 91.0 | 92.7 | 1.000 | 0.71 |

[a-b]Means within a row without a common superscript differ significantly ($P < 0.05$).

[1]PRSE - Polyphenol rich sugarcane extract.

[2]SEM – Pooled standard error of the mean (n = 6).

## Discussion

This study evaluated the effects of supplementing polyphenol-rich sugarcane extract (PRSE) in the drinking water of Shaver Brown laying hens, focusing on its impact on internal egg quality, antioxidant capacity, and production performance. The findings offer insights into

**Table 4. Effect of supplementing polyphenol-rich sugarcane extract in drinking water on the antioxidant properties of egg yolk.**

| Variable | PRSE[1] concentration (%) in drinking water | | SEM[2] | P value |
|---|---|---|---|---|
| | 0 | 0.05 | | |
| 2,2-diphenyl-1-picrylhydrazyl assay (mMTrolox Eq/g) | | | | |
| Week 45 | 72.2[a] | 45.6[b] | 6.152 | 0.02 |
| Week 47 | 51.7 | 51.6 | 1.767 | 0.98 |
| Week 49 | 47.7 | 62.7 | 4.561 | 0.10 |
| Ferric Reducing Antioxidant Power Assay (mMFe$^{2+}$ Eq/g) | | | | |
| Week 45 | 436052.0 | 425965.3 | 46671.39 | 0.92 |
| Week 47 | 430418.7 | 417682.0 | 29532.87 | 0.84 |
| Week 49 | 506055.3 | 474492.0 | 54280.62 | 0.79 |

[a-b]Means within a row without a common superscript differ significantly ($P < 0.05$).

[1]PRSE - Polyphenol rich sugarcane extract.

[2]SEM – Pooled standard error of the mean (n = 6).

how PRSE supplementation influences egg quality and antioxidant properties, while also highlighting the complexity of these interactions and the need for further research.

A significant observation was the reduction in yolk color at week 45 in the PRSE-supplemented group. This reduction is likely due to the oxidation of pigments such as xanthophylls and carotenoids, leading to a lighter yolk color. Such an outcome is undesirable as it suggests a potential decrease in the nutritional quality of the eggs and deviates from the deep yellow color preferred by consumers [21]. The return of yolk color to normal levels in subsequent weeks might indicate an initial disruption in the hens' antioxidant defense mechanisms due to PRSE introduction, followed by a gradual adaptation to the supplementation. The polyphenols in PRSE, while generally antioxidant, could interact with other dietary or metabolic components, leading to an overreaction of certain oxidative pathways, which may have caused a transient period of carotenoid oxidation. Additionally, if PRSE supplementation affected the absorption or transport of carotenoids into the yolk, this could also contribute to the observed reduction in yolk color [22]. The return of yolk color to normal levels in the following weeks suggests that the hens' antioxidant systems adapted to the presence of PRSE, enabling a more stable environment for carotenoid preservation. The transient nature of this effect raises questions about the role of PRSE in modulating oxidative processes within the egg yolk, potentially reducing the stability or bioavailability of yolk pigments [22]. Further research is needed to evaluate the effects of administering an optimized dose of PRSE over an extended period, focusing on observing additional improvements in egg quality parameters, including egg yolk color, as the results suggest a gradual adaptation to PRSE supplementation.

Despite the negative impact on yolk color, other internal egg quality parameters, including yolk height, thick albumen height, and Haugh units, were not significantly affected by PRSE supplementation, indicating that the overall internal quality of the eggs remained largely intact [23]. However, the reduction in thin albumen height observed at weeks 47 and 49 suggests that PRSE may have subtle effects on egg white consistency over time, warranting further investigation. Polyphenols can occasionally exhibit pro-oxidant properties depending on their structure, concentration, and environmental conditions. This pro-oxidant activity may have contributed to the reduced pigment stability in egg yolks, the observed decline in thin albumen height, and the lack of improvements in other egg quality parameters in the current study [24,25]. The study observed that yolk height, thick albumen height, and Haugh units remained unaffected by PRSE treatment, while a trend toward a reduction in thin albumen

height was noted at weeks 47 and 49. These results indicate that the effects of PRSE on egg quality are multifaceted and may influence different parameters in distinct ways. Notably, polyphenols have been reported to enhance magnum morphology, including improvements in epithelial height, cilia height, and magnum fold height, which are closely linked to albumen quality [26]. Consequently, further investigation into the effects of PRSE on these specific egg quality traits is warranted.

The antioxidant capacity of the egg yolk was also assessed, revealing a significant reduction in DPPH radical-scavenging activity at week 45 in the PRSE group. A lower DPPH value indicates an enhanced antioxidant capacity, suggesting that the polyphenols in PRSE were initially effective in improving the eggs' ability to neutralize free radicals [8]. However, this effect was not sustained over time, implying that the antioxidant benefits of PRSE may be short-lived or that the hens' metabolism adapted to the supplementation. As the hens' body systems became more efficient at managing oxidative stress, the external influence of PRSE on antioxidant activity may have lessened over time. Additionally, the polyphenols in PRSE are susceptible to changes in bioavailability over time, meaning their effectiveness could have been reduced as the study progressed [22]. Oxidative stress from other environmental factors could also have overwhelmed the antioxidant benefits provided by PRSE. The transient nature of this PRSE effect on antioxidant capacity indicates that sustained or optimized PRSE supplementation strategies may be required to preserve these benefits over time.

In contrast to previous studies that reported significant effects of polyphenol supplementation on egg antioxidant properties, this study's results may have been influenced by the approach used to assess antioxidant capacity. Previous research often employed different methodologies, such as measuring plasma and yolk malonaldehyde concentrations, GSH activity in plasma and yolk, and shelf life, which may provide more accurate insights into antioxidant properties [27]. Future studies should consider incorporating these parameters to obtain a more comprehensive understanding of polyphenol's effects on egg quality.

The supplementation of 0.05% PRSE in the drinking water did not lead to statistically significant improvements in hen-day egg production, hen-housed egg production, egg mass, or feed conversion ratio throughout the study period. Although there was a slight indication of significance in egg weight during week 45 (P = 0.06), these findings suggest that the concentration and method of PRSE administration used in this experiment may not have been sufficient to elicit notable benefits under the specific conditions tested. The trend toward a significant increase in egg weight at week 45 suggests potential benefits, which may become more pronounced with further research or optimized dosages. This effect is likely attributable to polyphenols enhancing feed efficiency in laying hens, thereby improving nutrient utilization and contributing to increased egg weight [15].

Polyphenols are widely recognized for their antioxidant, anti-inflammatory, antibacterial, anticancer, and antistress properties [28]. Previous studies have demonstrated their potential to improve health and productivity in laying hens, including reduced oxidative stress, enhanced egg quality, improved eggshell integrity, and positive effects on gut health [29,30]. However, the lack of significant effects observed in this study could be attributed to several factors.

First, the concentration of PRSE used (0.05%) may have been too low to produce measurable benefits. The efficacy of polyphenols is often dose-dependent [31], with higher concentrations sometimes required to observe significant outcomes. Conversely, excessively high doses of polyphenols can reduce nutrient digestibility and negatively affect weight gain [15], highlighting the importance of determining an optimal dosage. Additionally, the method of administering polyphenols through drinking water rather than feed could have influenced their bioavailability and absorption, potentially diminishing their effectiveness.

Polyphenols are partially absorbed in the small intestine, while the unabsorbed fraction passes into the hindgut, where it undergoes microbial metabolism by the gut microbiota. The resulting polyphenol metabolites exhibit higher bioavailability and influence the gut microbial composition [32]. The proportion of polyphenols reaching the hindgut may vary depending on whether they are administered through feed or water, due to differences in intestinal transit time [33]. Moreover, the composition of gut microbiota significantly influences polyphenol metabolism in the digestive tract. Consequently, variations in age, dietary composition, and environmental conditions among birds directly impact the gut microbial community, thereby altering polyphenol metabolism [34]. These variations could impact polyphenol bioavailability, potentially leading to differences in egg production and quality in laying hens.

Another consideration is the specific polyphenol profile of the sugarcane extract used in this study. Most previous studies that observed significant effects on egg production, quality, and antioxidant properties utilized polyphenols derived from other plant sources, such as tea, grape pomace, and other plant varieties [35–37]. The polyphenol content and type of polyphenols can vary significantly between these different sources, making it challenging to compare the results of this study with those of others. Sugarcane contains polyphenol compounds such as phenolic acids, flavonoids, and various glycosides [16]. In contrast, other plant-based polyphenol sources include a broader spectrum of compounds, such as anthocyanins, proanthocyanidins, and catechins [38,39]. These differences in polyphenol composition influence their metabolism, ultimately affecting the performance of birds. Additionally, polyphenols exist in both free and bound forms within the feed matrix, with insoluble-bound phenolics being associated with cell wall fibers like cellulose. Bound polyphenols have been shown to exhibit stronger antioxidant effects compared to their free counterparts. Thus, the source of polyphenols plays a critical role in determining their impact on egg production and quality in laying hens [40]. Some research has even found reduced egg quality parameters, including eggshell weight, eggshell strength, and albumen quality, with polyphenol supplementation derived from plant sources like tea polyphenols. These variations emphasize the complexity of polyphenol effects on laying hens and suggest that the specific polyphenols in sugarcane extract may interact differently with the metabolism of birds.

Furthermore, the age of the hens may play a crucial role in their response to polyphenol supplementation. Most studies that reported positive effects of polyphenols on egg quality and antioxidant properties involved a late laying period, which is above 50 weeks [35,36]. In contrast, studies that found negative or non-significant effects often involved an early laying period [41]. The age-related differences in metabolism and absorption of polyphenols could influence egg quality outcomes, suggesting that older hens might benefit more from polyphenol supplementation than younger ones. As hens age, physiological changes such as reduced nutrient absorption efficiency, altered metabolism, and hormonal shifts affect how they process and utilize nutrients, including polyphenols [42]. Aging also leads to increased oxidative stress and decreased immune function, making older hens more dependent on antioxidant support [43]. These factors likely explain why older hens may respond better to polyphenol supplementation, as they require more support to manage oxidative damage and maintain egg quality. The age of the hens at the start of this study (43 weeks) places them in a mature phase of production, where their physiological responses to dietary interventions might differ from those in early and late laying periods. This factor should be considered in future studies to optimize polyphenol supplementation strategies. Further, the six-week duration of this study might not have been long enough to observe the full effects of PRSE supplementation. Some benefits of polyphenols may take longer to manifest and extending the study duration could provide more comprehensive insights into the long-term impact of PRSE on laying hens.

Furthermore, strain-specific responses to polyphenol supplementation should also be considered, as different strains of commercial layers might exhibit varying sensitivities to dietary changes.

Given the mixed results of polyphenol supplementation reported in previous studies, it is essential to continue investigating the effects of PRSE and other polyphenol sources in laying hens. In the current study, a 0.05% concentration of PRSE was administered in drinking water, a dosage selected based on optimal performance outcomes observed in a previous broiler study utilizing the same product under similar conditions [44]. Future research should explore varying concentrations of PRSE, alternative administration methods, and longer study durations. Additionally, examining the effects of polyphenol supplementation across different strains and at various stages of the laying cycle could provide valuable insights into optimizing egg production and quality through dietary interventions.

## Conclusions

In conclusion, this study demonstrates that supplementing PRSE in the drinking water of Shaver Brown laying hens has limited effects on internal egg quality, antioxidant capacity, and production performance under the tested conditions. The transient reduction in yolk color and thin albumen height highlights the complexity of polyphenol interactions, which may be influenced by dosage, administration route, and hen age. This study lays the groundwork for understanding the effects of PRSE on commercial laying hens while emphasizing the need for further investigation. Future research should aim to optimize the dosage, duration of PRSE supplementation, administration method, and age-specific responses, and uncover the mechanisms underlying its influence on egg quality and antioxidant capacity. Addressing these aspects will help clarify the practical applications of PRSE in poultry nutrition and its potential to improve the productivity and health of commercial layers.

## Acknowledgments

Acknowledgment is granted to the staff of the Udaperadeniya Livestock Farm, Faculty of Agriculture of the University of Peradeniya for their valuable support in the execution of the animal trials.

## Author contributions

**Conceptualization:** Namalika D. Karunaratne, Ruvini Liyanage, Pabodha Weththasinghe, Barana C. Jayawardana, Anil Pushpakumara, Mathew Flavel.

**Data curation:** Namalika D. Karunaratne, Ruvini Liyanage, Eranga De Seram.

**Formal analysis:** Namalika D. Karunaratne, Sasmitha De Silva, Minoli Herath, Ruvini Liyanage, Barana C. Jayawardana, Eranga De Seram.

**Funding acquisition:** Namalika D. Karunaratne, Anil Pushpakumara, Mathew Flavel.

**Investigation:** Namalika D. Karunaratne, Sasmitha De Silva, Minoli Herath, Ruvini Liyanage, Pabodha Weththasinghe.

**Methodology:** Namalika D. Karunaratne, Sasmitha De Silva, Minoli Herath, Barana C. Jayawardana.

**Project administration:** Namalika D. Karunaratne, Pabodha Weththasinghe, Barana C. Jayawardana, Anil Pushpakumara, Mathew Flavel.

**Resources:** Ruvini Liyanage, Pabodha Weththasinghe, Barana C. Jayawardana.

**Software:** Eranga De Seram.

**Supervision:** Namalika D. Karunaratne, Mathew Flavel.

**Validation:** Namalika D. Karunaratne, Sasmitha De Silva, Minoli Herath, Ruvini Liyanage, Eranga De Seram, Anil Pushpakumara.

**Visualization:** Namalika D. Karunaratne, Sasmitha De Silva, Minoli Herath, Eranga De Seram, Anil Pushpakumara.

**Writing – original draft:** Namalika D. Karunaratne, Sasmitha De Silva, Minoli Herath.

**Writing – review & editing:** Namalika D. Karunaratne, Sasmitha De Silva, Minoli Herath, Ruvini Liyanage.

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
