## [Decision Letter · Decision Letter 0]

19 Nov 2024

PONE-D-24-45645Effects of supplementing a polyphenol-rich sugarcane extract through drinking water on egg production and quality of laying hensPLOS ONE

Dear Dr. Karunaratne,

Thank you for submitting your manuscript to PLOS ONE. After careful consideration, we feel that it has merit but does not fully meet PLOS ONE’s publication criteria as it currently stands. Therefore, we invite you to submit a revised version of the manuscript that addresses the points raised during the review process.

We look forward to receiving your revised manuscript.

Kind regards,

Agung Irawan

Academic Editor

PLOS ONE

Journal Requirements:

3.  Thank you for stating the following in the Competing Interest ssection:

“Mathew Flavel, a co-author of this manuscript is attached to the company that supplies the polyphenol product. However, he did not involve with manuscript writing, data analysis and interpretation.”

We note that one or more of the authors are employed by a commercial company

4. We note you have included a table to which you do not refer in the text of your manuscript. Please ensure that you refer to Table 3 & 4 in your text; if accepted, production will need this reference to link the reader to the Table.

Reviewers' comments:

Reviewer's Responses to Questions

**Comments to the Author**

1. Is the manuscript technically sound, and do the data support the conclusions?

Reviewer #1: Yes

Reviewer #2: Partly

Reviewer #3: Yes

2. Has the statistical analysis been performed appropriately and rigorously? 

Reviewer #1: Yes

Reviewer #2: No

Reviewer #3: Yes

3. Have the authors made all data underlying the findings in their manuscript fully available?

Reviewer #1: Yes

Reviewer #2: Yes

Reviewer #3: Yes

4. Is the manuscript presented in an intelligible fashion and written in standard English?

Reviewer #1: Yes

Reviewer #2: Yes

Reviewer #3: Yes

5. Review Comments to the Author

Reviewer #1: The objective of the manuscript was to evaluate whether the inclusion of a polyphenol-rich sugarcane extract (PRSE) in drinking water could improve egg production and the quality of commercial layers. This manuscript has been well written, the materials and methods used have met statistical standards. The writing of the results has described the results of the data obtained. In the discussion section, it is necessary to sharpen the scientific mechanism of polyphenols in relation to egg production and egg quality. In the discussion it is stated that the source of polyphenols greatly affects egg production and egg quality, what are the key things that cause polyphenols from sugarcane extract to have no impact on egg production and egg quality. Compare the polyphenol content given in this study with the polyphenol content from other reference sources that have a significant impact on egg production and egg quality in laying hens. How is the difference in polyphenol activity from different sources? Is the source of polyphenols related to the binding of polyphenols to other compounds?, so that it affects the performance of polyphenols when consumed by laying hens. Is there a difference in the mechanism of action of polyphenols on laying hens when given through drinking water and through supplementation in feed? This manuscript has not written a conclusion section, so a conclusion section needs to be added.

Reviewer #2: Dear Authors

I have carefully reviewed your manuscript and identified several areas for improvement to enhance the clarity and quality of the study. Overall, the introduction could benefit from a more cohesive structure, providing clearer background information on the poultry industry and the context of polyphenols and sugarcane. Additionally, further explanation is needed regarding the mechanisms through which PRSE affects the various variables analyzed, including the reduction in yolk color and its impact on antioxidant capacity. The results would be better presented in graphical form to facilitate comprehension, and additional details regarding the treatment duration and environmental conditions would help contextualize the findings. I also suggest further clarification of the rationale behind the chosen PRSE dose, as well as consideration of higher doses based on previous studies.

Reviewer #3: Comments: PONE-D-24-45645

1. The study on the inclusion of polyphenol-rich sugarcane extract (PRSE) in the drinking water of Shaver Brown hens provides valuable insights into its potential benefits and limitations. While the study highlights that PRSE supplementation did not significantly impact overall egg production metrics such as hen-day egg production, hen-housed egg production, egg weight, egg mass, or feed conversion ratio, it is important to delve deeper into the nuances of these findings. The trend towards significance in egg weight at week 45 suggests that there may be subtle benefits that could become more pronounced with further research or different dosages.

2. The significant reduction in yolk color during week 45 raises questions about the consistency of PRSE’s effects over time. This transient change indicates that while PRSE may influence certain egg quality parameters, these effects might not be sustained throughout the study period. It would be beneficial to explore the underlying mechanisms that cause these temporary changes and whether adjustments in dosage or duration of supplementation could lead to more consistent results.

3. The study also notes that yolk height, thick albumen height, and Haugh units were unaffected by PRSE treatment, while thin albumen height showed a trend towards reduction at weeks 47 and 49. These findings suggest that PRSE’s impact on egg quality is complex and may vary across different parameters. Further investigation into how PRSE interacts with the physiological processes involved in egg formation could provide a clearer picture of its potential benefits.

4. One of the most promising findings is the significant increase in antioxidant capacity in the PRSE group at week 45, as revealed by the DPPH assay. This indicates that PRSE has the potential to enhance the antioxidant status of hens, which could contribute to improved health and productivity. However, the temporary nature of this effect suggests that continuous or optimized supplementation strategies may be necessary to maintain these benefits.

5. It’s important to detail how water consumption was monitored during the study to ensure accurate dosing of PRSE.

6. The manuscript should elaborate on the potential mechanisms through which polyphenol-rich sugarcane extract (PRSE) influences egg production and quality.

7. Discussing the rationale behind the chosen dose of 0.05% PRSE and exploring the potential for dose optimization could add depth to the study.

8. Providing practical recommendations for poultry farmers based on the study findings would be valuable.

9. Overall, the study provides a foundation for understanding the effects of PRSE on commercial layers, but it also highlights the need for further research. Future studies should focus on optimizing the dosage and duration of PRSE supplementation, exploring its long-term effects, and elucidating the mechanisms behind its impact on egg quality and antioxidant capacity. By addressing these areas, researchers can better determine the practical applications of PRSE in poultry nutrition and its potential to enhance the productivity and health of commercial layers.

6. PLOS authors have the option to publish the peer review history of their article (what does this mean? ). If published, this will include your full peer review and any attached files.

**Do you want your identity to be public for this peer review?** For information about this choice, including consent withdrawal, please see our Privacy Policy .

Reviewer #1: **Yes: ** Cecep Hidayat (The National Research and Innovation Agency of Indonesia)

Reviewer #2: **Yes: ** Aan Andri Yano

Reviewer #3: **Yes: ** Ahmed A.A. Abdel-Wareth

---

## [Author Response · Author response to Decision Letter 1]

26 Nov 2024

The authors would like to extend their gratitude to the editor and reviewers for their valuable comments and suggestions to improve the quality of the manuscript.

Reviewer 1

Comment Response to reviewer

In the discussion section, it is necessary to sharpen the scientific mechanism of polyphenols in relation to egg production and egg quality. In the discussion it is stated that the source of polyphenols greatly affects egg production and egg quality, what are the key things that cause polyphenols from sugarcane extract to have no impact on egg production and egg quality. Compare the polyphenol content given in this study with the polyphenol content from other reference sources that have a significant impact on egg production and egg quality in laying hens.

Polyphenol content among different sources was compared in the discussion section. (L 317-321)

How is the difference in polyphenol activity from different sources? Is the source of polyphenols related to the binding of polyphenols to other compounds? so that it affects the performance of polyphenols when consumed by laying hens.

The bound form and free form of polyphenols and their effect on antioxidant activity were discussed as suggested. (L 321-325)

Is there a difference in the mechanism of action of polyphenols on laying hens when given through drinking water and through supplementation in feed?

The rate and extent of absorption and metabolism of polyphenols are different when administered through feed vs. drinking water according to the literature. Therefore, it was mentioned in the discussion section. (L 301-311)

This manuscript has not written a conclusion section, so a conclusion section needs to be added.

A conclusion paragraph was added. (L 357-368)

Reviewer 2

Comment Response to reviewer

Overall, the introduction could benefit from a more cohesive structure, providing clearer background information on the poultry industry and the context of polyphenols and sugarcane.

Introduction section was revised as suggested by adding information on poultry industry, polyphenols and sugarcane. (L 51-57, 73-75)

Additionally, further explanation is needed regarding the mechanisms through which PRSE affects the various variables analyzed, including the reduction in yolk color and its impact on antioxidant capacity.

Further explanation was included for PRSE effects on egg quality parameters. (L 244-247)

The results would be better presented in graphical form to facilitate comprehension, and additional details regarding the treatment duration and environmental conditions would help contextualize the findings.

Additional details on the treatment duration were included to enhance clarity. (L 104-105) Environmental conditions including temperature, humidity, and lighting durations were tried to maintain according to the Shaver Brown guidelines, and it was mentioned in the manuscript. However, exact values cannot be reached and maintained since it is an open-sided house. The authors consider the current presentation of results in the tables to be clear and concise, and therefore, no changes have been made to their format.

I also suggest further clarification of the rationale behind the chosen PRSE dose, as well as consideration of higher doses based on previous studies.

The justification for the chosen PRSE dose was added to the discussion section. (L 349-351)

Reviewer 3

Comment Response to reviewer

The study on the inclusion of polyphenol-rich sugarcane extract (PRSE) in the drinking water of Shaver Brown hens provides valuable insights into its potential benefits and limitations. While the study highlights that PRSE supplementation did not significantly impact overall egg production metrics such as hen-day egg production, hen-housed egg production, egg weight, egg mass, or feed conversion ratio, it is important to delve deeper into the nuances of these findings. The trend towards significance in egg weight at week 45 suggests that there may be subtle benefits that could become more pronounced with further research or different dosages.

The effect of PRSE on egg weight was further elaborated in the discussion section. (L 285-289)

The significant reduction in yolk color during week 45 raises questions about the consistency of PRSE’s effects over time. This transient change indicates that while PRSE may influence certain egg quality parameters, these effects might not be sustained throughout the study period. It would be beneficial to explore the underlying mechanisms that cause these temporary changes and whether adjustments in dosage or duration of supplementation could lead to more consistent results.

Further explanation was added related to the results of egg yolk colour. (L 244-247)

The study also notes that yolk height, thick albumen height, and Haugh units were unaffected by PRSE treatment, while thin albumen height showed a trend towards reduction at weeks 47 and 49. These findings suggest that PRSE’s impact on egg quality is complex and may vary across different parameters. Further investigation into how PRSE interacts with the physiological processes involved in egg formation could provide a clearer picture of its potential benefits.

Polyphenol's effect on egg formation physiological process was incorporated in the discussion section when clarifying the PRSE effects on internal egg quality parameters. (L 259-264)

One of the most promising findings is the significant increase in antioxidant capacity in the PRSE group at week 45, as revealed by the DPPH assay. This indicates that PRSE has the potential to enhance the antioxidant status of hens, which could contribute to improved health and productivity. However, the temporary nature of this effect suggests that continuous or optimized supplementation strategies may be necessary to maintain these benefits.

The authors agree with the reviewer's comment, and it was added to the discussion section. (L 270-272)

It’s important to detail how water consumption was monitored during the study to ensure accurate dosing of PRSE.

Water intake was not measured in the current study due to practical constraints, although the authors agree with the reviewer's comment on the importance of measuring water consumption. However, in a previous study evaluating broiler performance using the same PRSE in drinking water, water intake was measured and found to be not significantly different among treatment groups.

The manuscript should elaborate on the potential mechanisms through which polyphenol-rich sugarcane extract (PRSE) influences egg production and quality.

This comment was already addressed when adding sections to the discussion in relation to egg weight, yolk colour, and albumen quality parameters. (L 244-247, 253-264, 270-272, 285-289)

Discussing the rationale behind the chosen dose of 0.05% PRSE and exploring the potential for dose optimization could add depth to the study.

The justification for the chosen PRSE dose was added to the discussion section. (L 349-351)

Providing practical recommendations for poultry farmers based on the study findings would be valuable.

Due to the variability in the results, the authors are unable to provide practical recommendations at this stage and emphasize the need for further research, as outlined in the discussion and conclusion section. (L 362-368)

Overall, the study provides a foundation for understanding the effects of PRSE on commercial layers, but it also highlights the need for further research. Future studies should focus on optimizing the dosage and duration of PRSE supplementation, exploring its long-term effects, and elucidating the mechanisms behind its impact on egg quality and antioxidant capacity. By addressing these areas, researchers can better determine the practical applications of PRSE in poultry nutrition and its potential to enhance the productivity and health of commercial layers.

The authors agree with the reviewer comments mentioned under this point, and the information was incorporated into the conclusion section of the manuscript. (L 358-368)

---

## [Decision Letter · Decision Letter 1]

8 Dec 2024

PONE-D-24-45645R1Effects of supplementing a polyphenol-rich sugarcane extract through drinking water on egg production and quality of laying hensPLOS ONE

Dear Dr. Karunaratne,

Thank you for submitting your manuscript to PLOS ONE. After careful consideration, we feel that it has merit but does not fully meet PLOS ONE’s publication criteria as it currently stands. Therefore, we invite you to submit a revised version of the manuscript that addresses the points raised during the review process.

We look forward to receiving your revised manuscript.

Kind regards,

Agung Irawan

Academic Editor

PLOS ONE

Journal Requirements:

Additional Editor Comments:

According to authors' statement, Mathew Flavel should not have been listed as co-author as he/she did not involve in the manuscript preparation and writing and did not intellectually contribute to the manuscript. Please review the authorship requirement of the journal for more details. 

Please provide justification and/or rebuttal to the reviewer's comments for the revision. 

Reviewers' comments:

Reviewer's Responses to Questions

**Comments to the Author**

1. If the authors have adequately addressed your comments raised in a previous round of review and you feel that this manuscript is now acceptable for publication, you may indicate that here to bypass the “Comments to the Author” section, enter your conflict of interest statement in the “Confidential to Editor” section, and submit your "Accept" recommendation.

Reviewer #1: All comments have been addressed

Reviewer #2: (No Response)

Reviewer #3: All comments have been addressed

2. Is the manuscript technically sound, and do the data support the conclusions?

Reviewer #1: Yes

Reviewer #2: Yes

Reviewer #3: Yes

3. Has the statistical analysis been performed appropriately and rigorously? 

Reviewer #1: Yes

Reviewer #2: Yes

Reviewer #3: Yes

4. Have the authors made all data underlying the findings in their manuscript fully available?

Reviewer #1: Yes

Reviewer #2: Yes

Reviewer #3: Yes

5. Is the manuscript presented in an intelligible fashion and written in standard English?

Reviewer #1: Yes

Reviewer #2: Yes

Reviewer #3: Yes

6. Review Comments to the Author

Reviewer #1: In my opinion, this revised manuscript meets the following requirements:

1. The study presents the results of original research.

2. Results reported have not been published elsewhere.

3. Experiments, statistics, and other analyzes are performed to a high technical standard and are described in sufficient detail.

4. Conclusions are presented in an appropriate fashion and are supported by the data.

5. The article is presented in an intelligent fashion and is written in standard English.

6. The research meets all applicable standards for the ethics of experimentation and research integrity.

7. The article adheres to appropriate reporting guidelines and community standards for data availability.

Reviewer #2: Dear Authors

The revised manuscript provides a comprehensive evaluation of the effects of polyphenol-rich sugarcane extract (PRSE) supplementation on egg production, quality, and antioxidant capacity in laying hens. The authors have addressed most of the reviewers’ comments, including improving the introduction with relevant background on the poultry industry and the context of PRSE, justifying the chosen dosage, and elaborating on mechanisms influencing egg quality and antioxidant effects. The statistical analysis is rigorous, and the conclusions are well-supported by the data. However, there are areas that could benefit from further refinement, such as providing detailed environmental data (e.g., temperature and humidity), adding graphical representations to enhance the presentation of trends, and deepening the discussion on the mechanistic basis of PRSE’s effects on yolk color and antioxidant capacity. These improvements would further strengthen the manuscript's clarity and impact. Overall, the study is technically sound and provides valuable insights, with only minor revisions needed for enhanced readability and completeness.

Reviewer #3: (No Response)

7. PLOS authors have the option to publish the peer review history of their article (what does this mean? ). If published, this will include your full peer review and any attached files.

**Do you want your identity to be public for this peer review?** For information about this choice, including consent withdrawal, please see our Privacy Policy .

Reviewer #1: **Yes: ** Dr. Cecep Hidayat, National Research and Innovation Agency of Indonesia

Reviewer #2: **Yes: ** Aan Andri Yano

Reviewer #3: **Yes: ** Ahmed A.A. Abdel-Wareth

---

## [Author Response · Author response to Decision Letter 2]

12 Dec 2024

The authors would like to extend their gratitude to the editor and all three reviewers for their valuable comments and suggestions to improve the quality of the manuscript.

Reviewer 2

Reviewer comment Response to reviewer

The mention of a p-value of 0.06 could cause confusion and may be omitted (Line 39), please revise.

P value was removed (L 39).

The conclusion should be more prominent and clearer in its emphasis (Line 45).

The conclusion statement of the abstract was revised as suggested (L 45-46).

The connection between polyphenols and sugarcane should be explained more clearly (Line 57-64).

A statement was added to make the connection between polyphenols and sugarcane (L 75-76).

More specific information about the polyphenol content in sugarcane and its comparison to other grasses would strengthen the background (Line 65), please revise.

Specific information about sugarcane (compared to other crops) was added as background information (L 73-74).

The study should provide clearer explanations of previous research findings on sugarcane extract and its effects on livestock (Line 70). Please add them.

Explanation related to sugarcane extract effects on livestock, especially poultry was added to the introduction (L 83-84).

The method for selecting eggs for quality analysis could be clarified, particularly whether specific days were chosen for sampling based on production cycles (Line 136).

The study lasted for six weeks, with egg collection conducted every two weeks, resulting in three sampling days for egg quality analysis. Previous nutritional studies on egg quality, typically lasting around 20 weeks, have collected samples at four-week intervals. However, given the shorter duration of this trial, samples were collected more frequently at two-week intervals. The sampling schedule was also influenced by budgetary constraints, which limited the number of samples that could be analyzed.

Environmental conditions, such as temperature and humidity, should be specified as they may influence egg production and quality (Line 95).

The experiment was carried out in an open-sided poultry house, and therefore, the temperature and humidity could not be controlled. Further, temperature and humidity were different throughout the day, and also throughout the trial duration, and the authors could not monitor those parameters continuously since it is not an environmentally controlled house. However, the average temperature and humidity in that region were reported in the manuscript (L 123-125).

Trends in performance would be better illustrated in graphical form (Line 45-49).

Performance data were included as 2 graphs in the results section (Fig 1 and Fig 2; L 207-213).

Superscripts should be used in tables to indicate significant differences within rows (Table 2).

Superscripts were added to tables to indicate significant differences within rows (Table 3 and Table 4).

The reduction in yolk color at week 45 should be further explained with a discussion on the potential mechanisms behind it, particularly how PRSE may have caused oxidation or reduced stability of carotenoids (Line 219).

Potential mechanisms that can lead to the reduction of yolk colour were added to the discussion (L 255-261).

The loss of antioxidant effects over time should be discussed in more detail, particularly regarding whether the PRSE supplementation was able to sustain its antioxidant effects throughout the study period (Line 235).

This was discussed in more detail as suggested by the reviewer (L 289-294).

The age of the hens as a factor influencing the outcomes should be explored in more depth, with more references or evidence regarding age-specific responses to polyphenol supplementation (Line 279).

Explanations related to age-specific responses were added to the discussion (L 361-366).

The conclusion in the revised manuscript is a valuable summary of the study’s findings, but it could be enhanced by: Emphasizing the need for future research to focus on optimizing PRSE dosages, administration methods, and considering age-specific responses in poultry (Line 279).

The conclusion statement was revised by including the suggested points (L 394).

Editor

Editor’s comment Response to editor

According to authors' statement, Mathew Flavel should not have been listed as co-author as he/she did not involve in the manuscript preparation and writing and did not intellectually contribute to the manuscript. Please review the authorship requirement of the journal for more details. The authors intended to convey that Mathew Flavel did not impose his opinions on the study design or outcomes, so the statement was rephrased accordingly in the cover letter.

“The funder provided support in the form of salaries for authors [NDK], but did not inappropriately influence the study design, data collection and analysis, decision to publish, or preparation of the manuscript.”

Any changes to the reference list should be mentioned in the rebuttal letter that accompanies your revised manuscript Two new references were added to the discussion section, and therefore, the reference list was updated (L 516-521).

42. Batal AB, Parsons CM. Effects of age on nutrient digestibility in chicks fed different diets. Poultry Science. 2002; 81(3): 400-407.

43. Del Vesco, AP, Khatlab, AS, Goes ESR, Utsunomiya, KS, Vieira, JS, Oliveira Neto AR, et al. Age-related oxidative stress and antioxidant capacity in heat-stressed broilers. Animal. 2017; 11(10): 1783-1790.

---

## [Editor Report · Decision Letter 2]

26 Dec 2024

Effects of supplementing a polyphenol-rich sugarcane extract through drinking water on egg production and quality of laying hens

PONE-D-24-45645R2

Dear Dr. Karunaratne,

We’re pleased to inform you that your manuscript has been judged scientifically suitable for publication and will be formally accepted for publication once it meets all outstanding technical requirements.

Kind regards,

Agung Irawan

Academic Editor

PLOS ONE
---

## [Editor Report · Acceptance letter]

PONE-D-24-45645R2

PLOS ONE

Dear Dr. Karunaratne,

I'm pleased to inform you that your manuscript has been deemed suitable for publication in PLOS ONE. Congratulations! Your manuscript is now being handed over to our production team.

Kind regards,

on behalf of

Dr. Agung Irawan

Academic Editor

PLOS ONE